# Recapitulating the Vasculature Using Organ-On-Chip Technology

**DOI:** 10.3390/bioengineering7010017

**Published:** 2020-02-18

**Authors:** Andreas M.A.O. Pollet, Jaap M.J. den Toonder

**Affiliations:** 1Microsystems, Department of Mechanical Engineering, Eindhoven University of Technology, 5600MB Eindhoven, The Netherlands; a.m.a.o.pollet@tue.nl; 2Institute for Complex Molecular Systems, Eindhoven University of Technology, 5600MB Eindhoven, The Netherlands

**Keywords:** vasculature, organ-on-chip, microfabrication, microfluidics, angiogenesis, vasculogenesis

## Abstract

The development of Vasculature-on-Chip has progressed rapidly over the last decade and recently, a wealth of fabrication possibilities has emerged that can be used for engineering vessels on a chip. All these fabrication methods have their own advantages and disadvantages but, more importantly, the capability of recapitulating the in vivo vasculature differs greatly between them. The first part of this review discusses the biological background of the in vivo vasculature and all the associated processes. We then evaluate the biological relevance of different fabrication methods proposed for Vasculature-on-Chip, we indicate their possibilities and limitations, and we assess which fabrication methods are capable of recapitulating the intrinsic complexity of the vasculature. This review illustrates the complexity involved in developing in vitro vasculature and provides an overview of fabrication methods for Vasculature-on-Chip in relation to the biological relevance of such methods.

## 1. Introduction

Organ-on-Chip technology is becoming an important tool for biological and medical research because these small scale systems are able to recapitulate the complex micro environment of organs in a controlled manner. The basic design of these chips and the fact that only small amounts of material are needed for performing experiments make them a possible platform for high-throughput and cost-effective research. The field of Organ-on-Chip and, in particular, the field of Vasculature-on-Chip has been progressing quickly in the last decade. This development has been fuelled by several factors, such as the need to create a vascular network to overcome the diffusion limit in tissue engineering, enabling one to realize larger tissue constructs; recreating functional vessels for implantation; fundamental research into vessel formation; and creating in vitro drug testing models. Promising methods have emerged that can recapitulate vasculature in vitro, but these are mostly dedicated to the specific research question in mind. To validate the broader applicability of the different methods that are currently used, these models must be evaluated based on their capability to recapitulate the in vivo vasculature. Therefore, this review starts with a concise overview of the function and architecture of the vasculature, followed by a description of the in vivo processes involved in growth and remodelling of the vasculature. With this knowledge at hand, the different existing fabrication methods for Vasculature-on-Chip are then discussed, leading to an overview of the applicability and recapitulation potential of these models in a biological context. By summarising what is known about both in vivo and in vitro vasculature, we hope that researchers from both fields can benefit by gaining a comprehensive understanding and therefore, progress in the same direction. Our vision for the future is that Vasculature-on-Chip models and experiments will and can be designed in such a way that results can be better translated towards biology and medicine, leading to the next step in Organ-on-Chip technology. 

The present paper complements other available reviews with a different focus within the field of the vasculature; specifically, we would like to point the reader to reviews about tissue vascularisation [1], vascular permeability [2], and organ specific vasculature [3]. Most existing reviews focus on either the biological or technological aspect of Vasculature-on-Chip or on a specific application. With this review, we aim to bridge this gap and make readers aware of the biological background involved as well as the current technological capabilities, including an evaluation of the technical merits. 

## 2. Architecture and Function of the Vasculature 

The vasculature is the main network inside the body for transporting oxygen, nutrients, and signals. This network can roughly be divided into three parts: The arterial part that transports the blood from the heart to the tissue; the capillaries, which distribute the blood through a fine network within the tissue and exchange oxygen and nutrients with the tissue; and the venous part, which collects all the blood from the capillaries to transport it back to the heart. Figure 1 schematically depicts the architecture of these different parts and shows that they all have their own specific structure to fulfil their function, yet a common feature is that the entire vasculature is lined on the inside with endothelial cells. The main function of these endothelial cells is to create a continuous lining that keeps the blood separated from the tissue but still allows for control over the exchange of molecules across the vessel wall. Table 1 lists the characteristic dimensions, blood flow velocities, and pressures for the different parts of the vascular tree.

The arterial part of the vasculature can be further subdivided into the elastic artery, muscular artery, and arterioles. The elastic arteries are for instance the aorta and the main distributing branches closest to the heart [4,5,6]. These arteries are large (1–2.5 cm in diameter) with a thick wall (1 mm) and consist mostly of elastic tissue and smooth muscle cells [4,6]. The elastic tissue dampens the pulsatile flow of the beating heart in the aorta by functioning like a compliant tube (40–50 cm·s^−1^ flow velocity and 80–120 mmHg = 10.5–16 MPa pressure) [4,6]. Because of the dampening effect, the blood flow becomes less pulsatile and the pressure becomes lower further away from the heart. Relative to the elastic arteries, the muscular arteries have more smooth muscle cells and less elastic tissue, which results in stiffer vessels [4]. In contrast to the elastic artery, the smooth muscle cells in the muscular artery are able to contract and control blood flow by locally changing the inner diameter and thus, the resistance of the vessel, controlling the flow and pressure throughout the network [4,5,6]. The diameter of the muscular artery ranges from 1 cm to 0.3 mm with a wall thickness of around 1 mm [4]. These properties help to fulfil the main role of the muscular arteries as distributors of blood throughout the body. Arterioles are the smallest of all arteries (0.3 mm–10 µm in diameter) and for the smallest diameter, they only have a single layer of smooth muscle cells [4,5,6]. The main function of arterioles is to further distribute and control blood flow by opening or closing different parts of the network. In this way, perfusion can be controlled to direct oxygen and nutrients to the sites where these are needed the most. The main bypass route for controlling perfusion are arteriovenous shunts that circumvent the capillary bed by connecting the arteries directly with the veins [4,6]. 

The capillaries are the smallest of all vessels (5–10 µm in diameter) that only consist of an endothelial lining with a basement membrane and some pericytes as support [1,4,5,6]. The highly branched capillary network inside tissue is needed for overcoming the diffusion limit of oxygen and nutrients inside the tissue, providing every cell inside this tissue with nutrients and oxygen. The network structure is mostly based on metabolic demand and architecture, however all capillaries have a pressure close to that of the interstitial fluid and a low flow speed (5–0.03 cm·s^−1^). The low pressure and flow speed contribute, besides vessel wall permeability, to the ability for efficient exchange of solutes between blood and interstitial fluid. This exchange is further modified by having a different type of endothelial lining [4,6]. This lining can vary from a tightly connected continuous layer of endothelial cells to a fenestrated endothelium with pores going through the endothelial cells, or even a sinusoidal endothelium with openings between the endothelial cells and pores in the basement membrane. Continuous endothelium is the most abundant in the human body, but some specific organs require more permeable vessels for solute exchange, such as the kidney, intestine, and endocrine glands (fenestrated endothelium) or the liver, spleen, bone marrow, and lymph tissue (sinusoidal endothelium) [4,6,7].

The third part of the vascular system is the venous part, consisting of venules and veins. The venules (8–100 µm in diameter) are mainly collecting channels of the capillary bed. However, these venules play an important role in the case of infections as this is the primary site where leukocytes can migrate from the blood into the tissue [4,5,6]. The vessel wall of the venules thickens downstream with smooth muscle cells, but not as much as for their arterial counterparts. The veins (100 µm–2 cm in diameter) are the last collecting ducts towards the heart and have a wall consisting mostly of collagen and smooth muscle cells. Within these veins, valves, together with compression of surrounding tissue, direct blood flow back to the heart. Because of the lower pressure in the venous part, the vessel wall is thinner compared to the arterial vessels with a similar diameter. To prevent any further pressure loss during transport, the veins have a large diameter, resulting in a lower resistance and therefore enabling efficient blood transport. 

## 3. Biological Processes in the In Vivo Vasculature

In vivo vasculature is a dynamic system that constantly adapts to different biochemical and physical cues. During development and wound healing, the remodelling of the vasculature is most prominent and is hallmarked by the formation of new vasculature. However, even after the new vasculature has formed, it continues to change and mature towards a homeostatic situation. For the sake of clarity, all processes involved in the formation and remodelling of the vasculature are described separately, even though these processes can and mostly will take place simultaneously. Figure 2 schematically depicts all remodelling processes in the vasculature. 

### 3.1. Vasculogenesis

The first vessels inside the body are formed by a process called vasculogenesis, which is defined as forming a vasculature network de novo [4,5,8,9,10]. This process starts with angioblasts (endothelial precursor cells) differentiating at the edge of small blood islands (Figure 2A1) [4,5,11]. These precursor cells form small clumps of differentiated cells (endothelial cells) and start invading the tissue with small extensions from the main mass (Figure 2A2) [4,12]. These extensions of cells grow towards each other and form connections between the cell masses, forming a rudimentary network of endothelial cells (Figure 2A3) [4,8,9,10,11,12]. This network opens up for perfusion and is further stabilised via maturation (see maturation, below) (Figure 2A4) [4,5,8,10,11,12]. At this stage, the fate of the vessel is not yet determined but its structure resembles that of a capillary vessel (Figure 1) and during maturation, the vessel can adopt either an arterial or venous outcome [8,9,10,11]. Vasculogenesis occurs mostly during embryonic development but can also take place during wound healing, initiated by circulating endothelial precursor cells [5,9].

### 3.2. Angiogenesis

Angiogenesis is the formation of new vasculature from an existing vessel and is the main method in the (mature) body for creating new vessels. This process is triggered by a decrease in oxygen concentration within a tissue (hypoxia), resulting in higher levels of hypoxia inducible factors (HIF) inside a cell (Figure 2B1) [13]. This increased HIF leads to elevated expression of vascular endothelial growth factor (VEGF), one of the main angiogenic factors that activates endothelial cells [8,9,10,11,13,14,15]. VEGF expression creates a gradient opposing the oxygen gradient in the tissue that can be sensed by the endothelial cells to guide angiogenesis (Figure 2B1). When endothelial cells sense the increase in VEGF, they switch from a quiescent state into a pro-angiogenic state [9,11,15]. In this state, the endothelial cells dilate the vessels and loosen their cell-cell contacts (VE-Cadherin), disrupting the continuous lining of endothelial cells [9,15,16]. As a consequence, fibrinogen leaks out of the vessels into the surrounding matrix, providing a new temporary matrix suitable for migration [17,18]. The original matrix (basement membrane) is broken down by metalloproteases (MMP) expressed by endothelial cells [9,10,15,16,17,18]. The pericytes that normally stabilise the endothelial cells via N-Cadherin detach under the influence of angiopoietin 2 (ANG-2) expressed by endothelial cells [10,11,14,15]. Liberated endothelial cells start to migrate towards the VEGF source, forming so-called sprouts that consist of a leading tip cell followed by stalk cells [9,10,11,19]. The tip cell has filopodia (finger like structures) that sense the environment. Using these filopodia, it migrates through the extracellular matrix using cell–matrix connections (Integrin) (Figure 2B2) [9]. Existing extracellular matrix is degraded by MMPs, enabling migration through the matrix. The stalk cells follow and proliferate, staying connected to the tip cell and to the original vessel via VE-Cadherin [9,10]. The process of tip cell selection is controlled by Notch signalling which is a cell–cell signalling pathway for determining cell fate. The tip cell expresses the ligand delta-like-4 (DLL4) that signals the Notch-1 receptor on the stalk cell to suppress tip cell fate [9,10,11]. Stalk cells express mainly the ligand Jagged-1 that promotes the tip cell in adopting a tip cell fate and angiogenesis. The role of the tip cell can be handed over to a stalk cell if a stalk cell expresses a higher level of DLL-4. This also applies to branching as additional tip cells can emerge from an existing sprout [20]. The sprouts follow gradients of chemotactic factors like VEGF, Semaphorins, and Ephrins as well as physical forces from interstitial flow, strain, and matrix stiffness (Figure 2B3) [9,11,16,18,21,22,23,24]. During sprout formation, the inside of these sprouts open up, forming a lumen that can be perfused from the original vessel. Fibroblasts remodel the extracellular matrix, providing the new matrix that was degraded by MMPs [15,25]. This remodelling of the extracellular matrix also plays an important role in the guidance of the angiogenesis process. Different chemotactic molecules can be bound and released by the extracellular matrix, creating additional gradients and tuning the function of these molecules [25]. For instance, VEGF in its soluble form promotes vessel enlargement while the bound form induces vessel branching [9]. 

### 3.3. Intussusception

Besides sprouting, existing vessels can also split by intussusception or bridging (Figure 2E) [11,19,26]. In this process, the vessel is split in two smaller vessels by tissue pillars growing inwards into the lumen. A complete splitting can be achieved by a row of these tissue pillars fusing together, or an existing bifurcation can be moved upstream by a similar row of pillars [26]. The endothelial lining stays intact during the process because the cells are pushed inwards until they make contact with the other side. After this point, the tissue pillar can further expand, moving the newly formed vessels further apart. 

### 3.4. Anastomosis

Anastomosis is defined as the process of connecting sprouts or a sprout with an existing vessel. This starts by sprout growth towards the source of the angiogenic signal to provide perfusion to this region. To fulfil this task, directional flow is required through the network, replenishing oxygen and nutrients continuously. This is possible by the fusion of two sprouts into one continuous vessel making a connection between the arterial and venous side of the vasculature (Figure 2C1) [10,11,19]. Anastomosis can also occur for sprouts connecting to existing vessels, creating a higher branching network. The process of anastomosis is guided by the same gradients as in angiogenesis. Additional guidance for connecting both tip cells is obtained via strain gradients [18,24]. This gradient is created by the tip cells pulling on the fibres inside the extra cellular matrix straining these fibres. Fibres that are pulled from both sides experience the highest strain which guide tip cells to migrate towards each other. Once the two tip cells reach each other, the filopodia of both cells make contact and connect via VE-cadherin, stabilising the connection (Figure 2C2) [10,19]. This process can be aided by myeloid cells expressing Notch-1 but this is not required [10]. Anastomosis with existing vessels follows a similar route by making a connection between tip and vessel cells through VE-cadherin [19]. 

Once a stable connection is made, the surrounding matrix is further remodelled to provide a stable basement membrane for the endothelial cells to form a continuous lumen (Figure 2C3) [10,19]. The formation of the lumen can either be achieved by having pressure in one sprout pushing the lumen forwards through the sprout, or by formation of lumens at different points in the sprout that fuse [19]. Once perfusion is established, the oxygen concentration increases, resulting in a local decrease in VEGF. If required, further angiogenesis can occur depending on residual angiogenic ques being present. 

### 3.5. Lumen Formation

The formation of a lumen, as mentioned earlier, is a crucial process during angiogenesis to provide perfusion to the tissue. Lumen formation can occur via different routes, namely: fusion of vacuoles or repulsion of the membranes (Figure 2F) [10,11]. In the first case, small pockets of liquid (vacuoles) are released on the apical (will be luminal) side of the cell. These vacuoles, which are normally also present in the cell, fuse and thereby form an increasingly larger lumen until it connects with the main lumen. For the second route, negatively charged glycoproteins are expressed on the luminal side, creating a repulsion force between the two membranes of the endothelial cells. Subsequent remodelling of the internal cytoskeleton results in the formation of a lumen between the two endothelial cells. In both cases, the endothelial cells remain in contact via VE-cadherin at the edges, maintaining a continuous endothelial lining. 

### 3.6. Maturation

During the formation and remodelling of vessels, all components are in a highly dynamic and plastic state. Further maturation is required to form stable vessels that are able to withstand the flow and associated forces as well as to contain the blood inside the lumen. Therefore, the maturation process starts immediately after activation to recover from the initial angiogenic cue and to establish vessel integrity. The sprouting endothelial cells are initially not stable and need support from pericytes and extracellular matrix components to fulfil their function [24,25,27]. Maturation starts with the expression of platelet derived growth factor (PDGF) by endothelial cells, which results in chemotaxis of fibroblasts towards these endothelial cells (Figure 2D1) [9,10,11,17]. These fibroblasts will become pericytes that wrap around the vessels to further stabilise and control vessel function [17,22,27]. Fibroblasts switch to a pericyte phenotype when they make contact with the endothelial cells (Figure 2D3). This change is caused by the activation of Notch-3 on the fibroblasts by Jagged-1 on the endothelial cells [10]. Further cell–cell contact is established via N-cadherin, connecting the pericytes to the endothelial cells, stabilising the vessel wall. Pericytes express angiopoietin 1 (ANG-1) which causes the endothelial cells to tighten the cell–cell junctions with VE-cadherin, increasing vessel tightness and resulting in endothelial quiescence [9,10,11,17,22]. Besides stabilisation with pericytes, vessels also form a basement membrane that provides mechanical strength and promotes integration into the surrounding tissue (Figure 2D3 Top) [9]. The formation of new basement membrane by pericytes is promoted by transforming growth factor beta (TGF-β), released by both endothelial cells and pericytes [9,10,11]. This basement membrane provides structure and, via β1-integrin, polarity to the endothelial cells for lumen formation [11]. A fully stable vessel wall is formed when the endothelial cells have become quiescent, pericytes are in close contact and around endothelial cells, and a basement membrane is present (Figure 2D3 Bottom). This stable state is preserved by: Notch-3-Jagged-1 signalling between endothelial cells and pericytes, shear stress acting on the endothelial cells, and low levels of autocrine VEGF-B (VEGF isoform), ANG-1, and fibroblast growth factor (FGF) [9,10,15,17,19]. Cell–cell contact is maintained via VE-cadherin between endothelial cells and via N-Cadherin between endothelial cells and pericytes [9,10,11]. Cell–matrix contact is maintained by integrin between the cells and extracellular matrix [9,11]. 

### 3.7. Regression

Stable vessels that become redundant regress; this remodelling process is called vascular pruning (Figure 2G) [9,10]. The main determinant for regressing a vessel is if the vessel is still perfused or not. If a vessel is perfused, the endothelial cells experience shear stress that acts as a survival signal. However, if perfusion drops, and therefore, the shear stress decreases, the endothelial cells start regressing, closing off the vessel lumen [10]. This regression progresses up- and down-stream until the point where shear stress is present again. After regression, only an empty basement membrane is left at the original site and this membrane is remodelled to tissue-specific extracellular matrix over time. 

### 3.8. Tumour Angiogenesis

Besides being an essential process in tissue maintenance and development, angiogenesis also plays an important role during tumour growth and progression. The vasculature remodels to provide growing tumours with sufficient nutrients and to establish a route for metastasis. The difference between normal and tumour angiogenesis is the imbalance between pro- and anti-angiogenic signals, which results in abnormal vascular formation [14]. There is a plethora of cancers and each of them has its own specific method for creating vasculature via angiogenesis [9,14,15,28]. A complete overview is beyond the scope of this review, nevertheless since a large part of the vascular research is focussed on this topic, a general overview of possible modes is given. 

In general, there are two distinct ways of tumour progression: (1) hypoxia based, where the tumour vasculature uses the same method as normal angiogenesis, or (2) vascular co-option, where the tumour grows around existing vasculature [8,9,10,13,14,15,27,28,29]. In the first case, tumour growth increases the demand for nutrients and oxygen, resulting in an increase in HIF signalling and VEGF expression starting angiogenesis [13]. Because of the abnormally high and continuous expression levels of angiogenic factors in cancer, the vasculature becomes ill-defined: the vessel wall is unstable and remains highly permeable, architecture of the vessel is tortuous and irregularly shaped, and the density of the vasculature is increased [10,15,27,29]. Further tumour growth results in a large cellular mass that increases interstitial pressure, resulting in stagnation of the flow and a poorly perfused necrotic core in the tumour [10,29]. This results again in a higher expression of angiogenic factors, becoming a vicious circle and preventing the formation of stable and quiescent vasculature [10]. The highly dynamic tumours are able to mimic the functionality of the vasculature by having cancer cells take over the role and function of endothelial cells in the vasculature [9,20]. To circumvent the need for vasculature, some tumours co-opt with existing vasculature and grow along the vessels [8,9,14,15]. This is a viable method for growth until the demand of nutrients and oxygen becomes too large and hypoxia starts the angiogenic process [14,15]. 

The role of the vasculature is also important in metastasis; the process where cancer cells from a primary tumour can travel via the vasculature to different regions inside the body and form new tumours. The permeability and functioning of the vessel wall plays a major role in this process—in particular, the coverage of pericytes is important [27]. Complete coverage by pericytes in tumour vasculature provides stability to the vessels and increases functionality, resulting in tumour growth and progression [27]. However, poor pericyte coverage gives rise to decreased functionality and higher levels of tumour hypoxia but an increased chance in tumour cell intravasation into the vasculature and possible metastasis [27].

### 3.9. Recapitulation

A list can be made of the components that have to be present based on the architecture of in vivo vessels, given in Table 2. These are, most importantly, endothelial cells that form a lining inside the lumen. These endothelial cells need support from pericytes and matrix proteins to form a stable and fully developed vessel. The process of vessel formation and homeostasis is controlled by numerous factors such as: growth factors, extracellular components, and physical ques. All of these affect the function of the cell and, depending on cell type, result in activation or inhibition of processes within the cell. Combining all the different players and influencers results in a complex orchestrated process steering the formation of the vasculature. This complexity is reflected in the shape of the vascular network and the layered structure of the vessel itself. In the next chapter, we will discuss different fabrication methods and look into how they are capable of controlling all these individual components needed to form a vascular network as found in vivo. 

## 4. Current Models for Recapitulating Vasculature-On-Chip

Research in the field of angiogenesis and the vasculature could greatly benefit from microfluidic technology that enables Vasculature-on-Chip and from the methods associated with it. This is because microfluidic technology makes it possible to manipulate physics, chemistry, and biology at the scales of the vasculature, creating a controllable environment mimicking the in vivo situation. One of the strengths, in particular, of Vasculature-on-Chip is that individual factors can be well controlled, for example, cells, geometry, flow, matrix properties, etc. and can be switched on or off or be modified. This enables one to unravel the roles of these factors, individual or combined [2,16,30,31,32,33,34,35]. However, it is important to ensure that the chosen method is best suited for the given research question while keeping the biological relevance in mind. To assess this for Vasculature-on-Chip, we summarize the different approaches by dividing them into general categories based on the fabrication method used and discuss the different variations as well as possible applications (Figure 3). We illustrate the approaches with concrete examples that are typical for the methods discussed, but we are far from exhaustive in doing so [1,16,30,36,37].

### 4.1. Templating

Templating covers all methods that use a template that is removed after casting of matrix material. After removal of the template, a lumen is created that can be covered with endothelium by seeding endothelial cells inside this lumen. One of the most commonly used methods is the casting of matrix material around small diameter needles or fibres that are removed after solidification of the matrix [38,39,40,41,42]. One of the main limitations is the minimal lumen diameter that can still be successfully seeded with cells: the reported limit is around 100µm, but new protocols have emerged that can lower this limit [43,44]. Based on the choice of materials for the matrix and cells, different research questions can already be answered with this simple model. Most common are studies on vascular permeability, angiogenesis, and the influence of flow and shear stress [39,40,45,46]. The main limitation, however, remains that only simple architectures can be fabricated. Single straight channels can be fabricated with needles and additional techniques with fibres or using viscous finger patterning can achieve simple bifurcations or stenosis, but these remain relatively simple compared to the in vivo vasculature [39,47,48]. 

### 4.2. Layer-By-Layer Composition

To further increase the complexity of the networks and to be able to make more intricate designs, the use of layer-by-layer fabrication has been applied. The layer can be made either from biodegradable materials [49], dissolvable polymers [50], or by crosslinking the matrix in patterns by, for example, using photolithography [51]. One of the main challenges with this technique is connecting the layers in such a way that they are properly aligned without any seams. Misalignment and seams will lead to discontinuities in the wall, disturbing the flow and adhesion of endothelium. The layer-by-layer fabrication also results in predominantly square shaped channels that do not resemble the in vivo round lumen shape that is critical for normal functioning [52]. Either high resolution techniques are required to create round channels on the micrometre scale of the vasculature or post processing must be applied to reshape the square channels into round ones. These types of fabrication methods are typically used for creating lattice networks to provide enough nutrients and oxygen to large tissues [49,51]. 

### 4.3. D Printing Sacrificial Template

To overcome the difficulties in aligning and creating round channels using layer-by-layer composition, 3D printing of a sacrificial template can provide an alternative. Because of the inherent round shape and alignment of the printed structures, reproducible networks can be created. One of the more simple methods is templating a sacrificial structure on top of the matrix and after printing, covering the structure with additional matrix, forming a channel [53,54]. This method results in easy to produce structures requiring little optimisation in printing parameters, but limits the design freedom to in-plane structures. The design freedom can be increased by printing inside the matrix, but this requires specific properties for both matrix and printing ink, limiting the choice of extracellular matrix [55]. To keep the freedom of matrix choice, casting around printed structures is one of the possibilities. However, this increases the demands for the printing ink to be self-supporting when printing freely suspended structures in space [56]. Several materials are available for printing these freestanding structures, with and without the help of support material, resulting in complex 3D networks that can be seeded with endothelial cells in a large range of matrices [57,58,59]. The limiting factor remains the minimum diameter that can be fabricated and seeded. 3D printing of sacrificial templates is mostly used for creating perfusable networks supporting large tissues and investigating the remodelling of the vasculature [57,58,59].

### 4.4. Laser Ablation

To work around the challenges of 3D printing, the reverse order can be followed in fabrication, starting with a block of matrix and removing material where the channels that mimic the vessels should be. Currently, laser ablation is one of the methods capable of doing this in a controllable way with high resolution in different matrix materials [60,61]. This results in complex networks that mimic the in vivo vasculature in size and geometry and that can be biologically active by performing the ablation in cell loaded matrices [62]. The difficulties with this method are: control over cell location within the matrix and thus the architecture, the relatively long fabrication time, and limited size due to the optical working distance. Laser ablation is especially useful for creating accurate copies of known vasculature networks or editing existing networks on the spot [61,62]. It is, therefore, a powerful tool for investigating flow and distribution behaviour in a complex network [62]. 

### 4.5. D Printing Cell/Matrix Mixture

By printing a cell laden matrix ink, control over cell location is increased and multi-layered structures can be fabricated, containing different components [63,64]. The same methodology can be followed as with 3D printing of artificial templates, but in this case, printing layered structures with a different matrix and cell composition that will form the vascular network [63,64,65]. An alternative to layer-by-layer printing is directly printing vessels by coaxial extrusion, using a smaller diameter nozzle inside another nozzle to dual extrude two materials in a fibre and sleeve manner. This coaxial extrusion creates multi-layered tube structures that can be arranged in specific patterns [66,67]. Despite the inability of fabricating in-plane bifurcations with this method, junctions can still be obtained via the remodelling of the matrix by the cells, connecting adjacent vessels [66]. So, in general, 3D printing cell laden inks offers less control over the geometry, but the in vivo multi-layered structure can be better mimicked. These 3D printing methods use a cell/matrix mixture that is biocompatible and have the freedom in vessel network design. Therefore, this method is mostly used for tissue engineering of vasculature or to achieve perfusable organs [63,64].

### 4.6. Angiogenesis-Based Platforms

Creating a Vasculature-on-Chip can also be done by triggering the biological activity of the vessels into forming new vessels and vasculature, rather than engineering the network using technical approaches. This results in a self-assembling network with less control over the geometrical details but offering the possibility to create more complex and potentially biologically more relevant structures. For systematic research in angiogenesis, several platforms have been created that enable control over the cell and matrix composition, as well as chemotactic and physical cues [36,37]. A simple version of a platform is based on needle templating of two parallel channels (see above), one acting as the vessel and the other as the source for the angiogenic factors [46,68]. Optionally, a single vessel can be fabricated and perfused with angiogenic factors [69]. These models can capture the effect of global gradients, but they do not capture local gradients as present in vivo. To further investigate the effect of gradients on sprouting, microfluidic devices have been designed that can create several local gradients in a highly predictable way. For example, more complex gradients and gradient combinations have been generated by including sources on both sides of the vessel channel [70]. Orthogonal gradients have been produced by having perpendicularly placed sources [71]. These types of experiments can give insight into the signalling cross talk between angiogenic factors and their outcome. One of these methods resulted in the finding that fibroblasts play a crucial role in the formation of the vascular networks by releasing growth factors into the environment that stabilise networks [72], and that pericytes are required for the formation of stable networks [73]. This platform consisted of three parallel gel chambers separated by two medium channels [74]. By loading the central gel chamber with endothelial cells, different gradients can be created either by adding growth factors to the medium channels or adding fibroblasts or other cell types in the side gel chambers. A similar chip design was used for investigating the effects of physical cues on sprouting; in this case, interstitial flow was generated that promoted sprouting against the direction of the flow [21]. These type of systems have also been used in the research on tumour vasculature and the effect of different drugs on the vasculature and tumour progression [75,76,77,78,79]. 

### 4.7. Vasculogenesis-Based Platforms

The same chip design can also be used to trigger vasculogenesis; in that case, instead of seeding endothelial cells next to a matrix, the matrix is loaded with endothelial cells and fibroblasts [74,80] or even together with organ specific cells (e.g., neuronal) [81] and tumour cells [82]. These systems rely more heavily on the biological components performing as normally happens in vivo, resulting in a more truthful network but with less control over individual components. These platforms have a more top-down approach in conducting research since a number of biological processes are already used for forming the networks. Despite the drawback of having less control, robust networks can be created with these platforms that enable the investigation of several components, such as flow and paracrine signalling between cells [83,84]. These systems can be easily up-scaled with many parallel individual chambers under similar conditions [85,86], which makes these systems suitable for research in tumour progression and possible drug targets [87]. Because of their good in vivo mimicking capabilities, these vasculogenesis platforms can be a viable candidate for drug testing and screening. Furthermore, these platforms can give insight in the vascular remodelling under specific conditions, finding new factors that influence network formation.

### 4.8. Summary

As has been described above, each of the platforms has its own specific strengths and limitations (see Table 3 for an overview). Choosing a platform that is suitable for answering a specific research question depends on what variable needs to be controlled to address the question. Platforms that are easy in setup and use can already give a lot of information, as long as all the important components and influences are taken into account. More advanced platforms are capable of implementing a larger number of variables and controlling these independently. However, the results can be difficult to interpret and additional experiments could be required to validate these results. In general, the three key components (endothelial cells, extracellular matrix, and fibroblasts) have to be taken into account to faithfully approach the in vivo situation. All the platforms discussed earlier have the possibility to implement these components, but not to the same extent. Therefore, control experiments need to be carried out to validate the use of a different component than is normally found in vivo. Picking a possible platform asks for comprehensive knowledge of the process under investigation and needs checking if all requirements are met. If needed, additional experiments or reflection with literature are required to justify and validate the use of a specific platform.

## 5. Conclusions

Based on the information we have discussed regarding the vasculature and considering all the different components that are crucial for mimicking the in vivo vasculature truthfully, a list of criteria (Table 1, Table 2 and Appendix A) can be derived that have to be met for conducting relevant research so that Vasculature-on-Chip results can be related to the in vivo situation. One example of a pitfall could be the investigation of angiogenesis or permeability without pericytes, or in the absence of correct basement membrane, which can result in misinterpretations or lead to contradictory results when compared to in vivo studies. Besides the basic components required, the type of method or device used can have a large impact on the outcome of experiments. For instance, not having control over the interstitial flow can already result in different sprouting behaviour [21] or in false gradients [71]. Choosing a suitable system depends on the process and effect that is targeted in the research; a practical approach is to first generate an understanding of the in vivo process and deduct what the main factors are, as has been outlined in this review. Based on this understanding, a design can be made with the ability to independently control these parameters. Combining the strengths of biology, medicine, and engineering can result in platforms that are able to systematically find missing links and gaps in the current knowledge. 

## 6. Outlook

For this approach to be successful, biology and medicine should indicate what is expected from the engineered systems and what checks should be made to validate the model system. On the other hand, the systems should convince the fields of biology and medicine of the relevance of the system to the general understanding of the vasculature and the advantage of systematically unravelling the role of different factors. In conclusion: one should be critical about the capabilities and limitations of the Vasculature-on-Chip systems that are available and the design and interrogation of Vasculature-on-Chip should always be guided by the specific research question in mind. By keeping a critical eye on the strengths and limitations of the current systems, next generation Vasculature-on-Chip systems can enable the investigation of even more complex questions by more truthfully mimicking the in vivo environment.

## Figures and Tables

**Figure 1 bioengineering-07-00017-f001:**
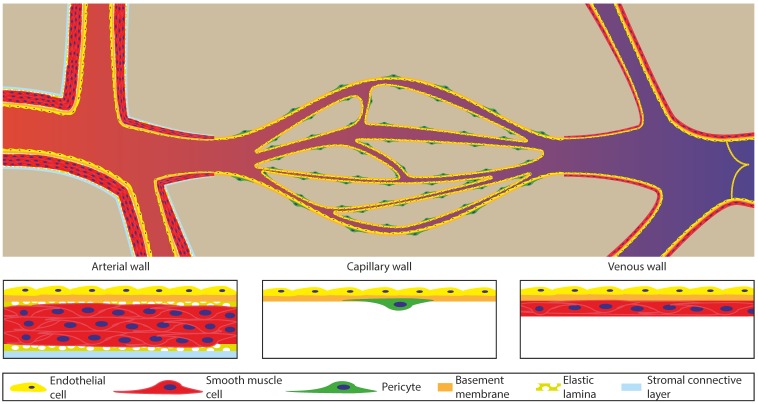
Schematic overview of the vasculature tree. Left: the arteries have large diameter vessels with a thick layer of smooth muscle cells. They transport blood from the heart and narrow down to transit to the capillary bed (middle) which bifurcates into smaller and closely spaced vessels to increase the exchange of oxygen and nutrients with the tissue and to decrease the required diffusion distance within the tissue. The capillary vessels consist mostly of only endothelial cells with a thin basement membrane supported by pericytes. The vessels bundle back together to form the venous part of the vasculature (right), which are again thicker and have a layer of smooth muscle cells, however not as abundant as in the arterial part. The venous system also has valves inside the lumen to prevent backflow and pumping action by the muscles around it, and it finally leads the blood back to the heart. Image inspired by [5,6].

**Figure 2 bioengineering-07-00017-f002:**
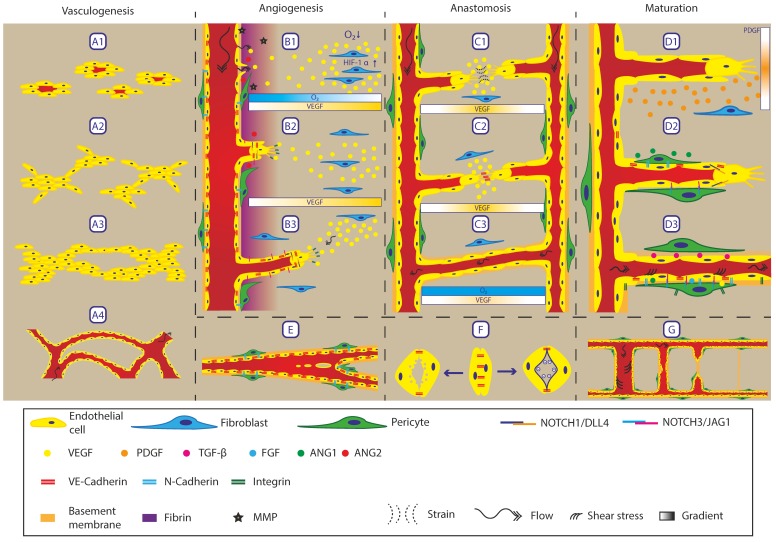
Overview of processes involving remodelling of the vasculature. **Vasculogenesis** is the process of forming de novo vasculature. *A1* The process starts with endothelial precursor cells scattered throughout the tissue. *A2* These precursor cells form clumps or islands of endothelial cells, *A3* which grow and form a primitive network structure. *A4* Upon full connection, the network opens up for perfusion, forming a primitive plexus. **Angiogenesis** is the process of forming new vasculature from an existing vessel. *B1* If oxygen concentration decreases, cells start to express VEGF via HIF-1α signalling which creates a gradient. When the vessel wall senses the increase in VEGF, the VE-Cadherin connections between endothelial cells loosen up, resulting in leakage of fibrin from the blood plasma into the surrounding matrix. Together with the expression and activation of MMPs, the matrix is remodelled to form a temporary matrix suitable for migration and sprouting. Sprouting endothelial cells release ANG2 upon which pericytes detach from the endothelial cells by disconnecting the N-Cadherin junctions. *B2* The initial VEGF gradient results in the selection of tip cells, which use integrins to migrate through the matrix and express DLL4 to inhibit neighbouring cells, adopting a tip cell phenotype via Notch1. Via this Notch signalling cascade, a single tip cell is maintained followed by stalk cells expressing Jagged1. *B3* Sprouts follow a gradient of VEGF, Semaphorins and Ephrins. Besides chemotactic migration, sprouts also migrate into the direction of interstitial flow, acting as a mechanical guidance. Due to Notch signalling within the sprout, following stalk cells stay connected to the tip cells, forming a continuous train of cells which opens up. *E* Besides sprouting, intussusception (or splitting angiogenesis) is also a method for forming new blood vessels. This process involves opposing endothelial cells making contact through the lumen of the blood vessel. After this, the vessel wall is remodelled until it is fully developed, resulting in a tissue pillar through the blood vessel. If this process is further promoted, a complete vessel can be split in two smaller vessels or bifurcations can be progressed into the vessel. *C1*
**Anastomosis** is guided via gradients, resulting in chemotaxis. Besides this gradient, the tip cells pull on the matrix to be able to migrate forward. This pulling results in fibres being strained between the two tip cells, forming a strain gradient that forms a guide for the tip cells to migrate towards each other and anastomose. *C2* Upon contact between two tip cells, the filopodia form a connection via VE-Cadherin, stabilising the connection. This process is guided by myeloid cells expressing Notch1. *C3* When the lumen of both sprouts are connected, the oxygen and nutrient levels increase, again reducing the expression of VEGF. *F* The formation of lumens can happen by two possible mechanisms. Left: Vacuoles are released from the endothelial cells towards the basal side of the forming lumen. This opens up a central cavity that can be perfused upon connection. Right: Glycoproteins that are negatively charged are present on the membrane at the basal side. Based on charge repulsion, the two membranes move away from each other, forming a lumen. **Maturation:**
*D1* during angiogenesis, sprouting endothelial cells express PDGF, which recruits pericytes to stabilise the vessel wall. The expression of PDGF forms a chemotactic gradient for fibroblasts to migrate towards the sprout and adopt a pericytes phenotype. *D2* When fibroblasts connect to the sprouts, they adopt a pericyte phenotype. Based on this switch, N-Cadherin is expressed, connecting the pericytes to the endothelial cells and stabilising the vessel wall. Expression of ANG1 by the pericytes results in clustering at the cell–cell junctions to increase vessel tightness and results in endothelial cell quiescence. *D3* Top: Expression of TGF-promotes ECM production and proliferation by pericytes as well as inducing a pericyte phenotype. Bottom: Fully stable and mature vessel wall. Pericytes adopt a phalanx state around the endothelial lumen. Low concentrations of VEGF, ANG1, and FGF are required for survival and stabilisation. Notch-3-JAG-1 signalling is required for survival and so are shear stresses, resulting from flow, acting on the endothelial cells. Endothelial cells are tightly connected together via VE-Cadherin and to pericytes via N-Cadherin; both endothelial cells and pericytes connect to the ECM via integrins. *G* Regression of vessels also occurs when perfusion is no longer required. The process starts by a reduced flow through the vessel resulting in a lower shear stress level. This low shear stress acts as a cue for endothelial cells to retract from the vessel and to close off the lumen. After complete retraction of the vessel, only the basement membrane is left, which will be remodelled over time to form a normal tissue ECM. Image inspired by [9,10,11].

**Figure 3 bioengineering-07-00017-f003:**
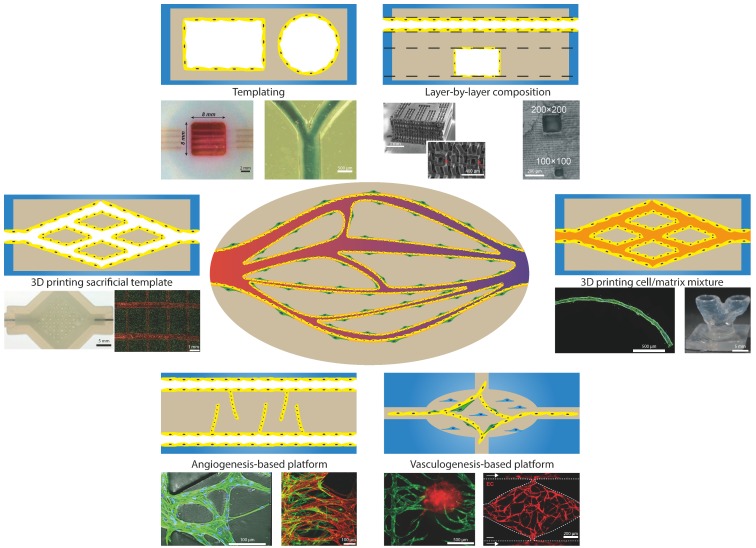
Different methodologies for recreating the Vasculature-on-Chip exist. These different methods can be divided roughly into the following groups. **Templating**: is based on having a template structure which can be removed after adding a matrix material with or without cells. This normally is done using a needle or fibre for creating circular channels or soft lithography for mostly square channels. Examples taken from: left [41] (scale bar 2 mm), right [47] (scale bar 500 µm). **Layer-by-layer composition**: Based on lithography processes, multiple layers with specific designs can be stacked upon each other, forming a complex 3D network. The main challenge is being able to properly align and bond multiple layers to achieve one connected network. Examples taken from: left [49] (scale bar top: 1 mm, bottom: 400 µm), right [51] (scale bar 200 µm). **3D printing sacrificial template**: This method is the next level beyond templating; more complex structures can be 3D printed and the desired matrix material can be casted around these structures. After solidification of the matrix, the printed structure is removed by dissolving, melting, or chemical breakdown, leaving an open structure. Examples taken from: left [57] (scale bar 5 mm), right [59] (scale bar 1 mm). **3D printing cell/matrix mixture**: By changing from a template to directly printing cells and matrix, more design freedom is obtained. However, the process becomes more complex and the printed networks are less accurate. Examples taken from: left [67] (scale bar 500 µm), right [64] (scale bar 5 mm). **Angiogenesis-based platforms**: This method employs angiogenesis from an existing vessel or an endothelial layer into a matrix, either by a chemotactic gradient or by activating angiogenesis in the endothelial layer. This method is able to recapitulate the sprouting and angiogenesis process to a large extent, however forming networks that resemble an in vivo vascular network remains challenging. Examples taken from: left [72] (scale bar 100 µm), right [73] (scale bar 100 µm). **Vasculogenesis-based platforms**: Here the biological process is stimulated to form the networks. By giving the right initial conditions (Endothelial cells, fibroblasts, and Fibrin in most cases, as well as flow), the mixture of cells and matrix is promoted to form vascular networks similar to in vivo ones. The control over this process is worse than in other methods but it can be steered by understanding the biological processes at play. Examples taken from: left [79] (scale bar 500 µm), right [87] (scale bar 200 µm). Images based on [1,41,47,49,51,57,59,64,67,72,73,79,87].

**Table 1 bioengineering-07-00017-t001:** List of characteristic dimensions, blood flow velocities, and pressures for all the different parts of the vascular tree.

	Elastic Arteries	Muscular Arteries	Arterioles	Capillaries	Venule	Vein
Diameter	2.5–1 cm	0.3 mm–1 cm	300–10 µm	10–5 µm	8–100 µm	100 µm–2 cm
Wall thickness	1 mm	1 mm	6 µm	0.5 µm	1 µm	0.5 mm
Pressure (S: Systolic, D: Diastolic)	120 S/90 D mm Hg	110 S/80 D mm Hg	80 S/60 D mm Hg	30 mm Hg	15 mm Hg	10 mm Hg
Blood flow velocity	50–45 cm·s^−1^	45–20 cm·s^−1^	20–5 cm·s^−1^	5–0.03 cm·s^−1^	5–10 cm·s^−1^	10–30 cm·s^−1^
Area of the vascular bed	2.5 cm^2^	250 cm^2^	2500 cm^2^	4500 cm^2^	3500 cm^2^	1000 cm^2^

**Table 2 bioengineering-07-00017-t002:** Components and factors that play a role in the vasculature.

Cells	Growth Factors	Extracellular Components	Physical Factors
Endothelial cells	Vascular Endothelial Growth Factor (VEGF)	Integrins	Oxygen Concentration
Pericytes	Platelet Derived Growth Factor (PDGF)	Matrix Metalloproteinase (MMP)	Interstitial Flow
Fibroblasts	Transforming Growth Factor β (TGF-β)	Fibrinogen	Shear Stress
	Fibroblast Growth Factor (FGF)	Collagen	Matrix Stiffness
	Angiopoietin signalling (ANG/TIE)	Laminin	Strain
	Notch signalling	Proteoglycans	

**Table 3 bioengineering-07-00017-t003:** General characteristics of the different fabrication methods.

Method	Biological Applications	Advantage	Limitation	In Vivo Recapitulation Strength
Templating	Permeability, angiogenesis, physical factors	Ease of use, round channel geometry	Minimal diameter, simple architecture	Low
Layer-by-layer composition	Perfusion, permeability	Large 3D networks	Alignment, channel geometry	Low
3D printing sacrificial template	Perfusion, remodelling	Round channel geometry	Complex print planning required	Medium
Laser ablation	Perfusion	High resolution and control	Needed equipment, fabrication time	Medium
3D printing cell/matrix mixture	Tissue engineering, perfusion	Biologically active	Less control over geometry, complex printing setup	High
Angiogenesis-based platforms	Angiogenesis, remodelling, perfusion, permeability	Close to in vivo vessels in anatomy and function	Less control over geometry, results depending on configuration and not always directly translational to in vivo	High
Vasculogenesis-based platforms	Vasculogenesis, remodelling, perfusion, permeability	Close to in vivo vessels in anatomy and function	Complex concert of factors needed, making individual components hard to investigate	High

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
