# Peer review of "Recapitulating the Vasculature Using Organ-On-Chip Technology"

_bioengineering, 2020, doi:10.3390/bioengineering7010017_

Round 1

Reviewer 1 Report

The manuscript presents an overview on vascular-on-chip technology. The manuscript is well written explaining the basic biological concepts and also the recent development of the lab-on chip technologies in the field. I have only two remarks:

-sections 4.1 to 4.5 describe the methods for fabrication while section 4.6 and 4.7 described the applications, maybe is better to separate in two different chapters

- the fabrication methods must be summarised in a table for a better understanding

Reviewer 2 Report

              In this paper, the author summarizes the architecture of the vasculature, biological process during the vascular formation, then discusses the engineering approaches to reconstruct the blood vessel in vitro. Although the paper is well written, I suggest the author considers several points to be revised.

Comment 1.

              As the author discusses in the introduction, because the research field to reconstruct a vascular network has been quickly progressed in this decade, a number of review papers focused on the topic. For example, Biomaterials, 35, 7308 (2014), Lab Chip, 15, 4242 (2015), Lab Chip 18, 2686 (2018), etc.. Especially, the Biomaterials paper has a similar table of contents discussed in this manuscript. I suggest the author refers these previous reviews and discusses why the readers should check the author's review in the introduction (the progress within a few years or categorizing the technical merits?).

Comment 2.

              Some important papers are missing in the manuscript. For example, Microvascular Research, 71 185 (2006) for 4.1 section, Lab Chip, 13, 1489 (2013) for 4.6 and 4.7 sections and others. If possible, add these papers to the manuscript.
